

# SSREnricher: a computational approach for large-scale identification of polymorphic microsatellites based on comparative transcriptome analysis

Wei Luo*, Qing Wu*, Lan Yang, Pengyu Chen, Siqi Yang, Tianzhu Wang, Yan Wang and Zongjun Du

College of Animal Science and Technology, Sichuan Agricultural University, Chengdu, Sichuan, China
* These authors contributed equally to this work.

## ABSTRACT

Microsatellite (SSR) markers are the most popular markers for genetic analyses and molecular selective breeding in plants and animals. However, the currently available methods to develop SSRs are relatively time-consuming and expensive. One of the most factors is low frequency of polymorphic SSRs. In this study, we developed a software, SSREnricher, which composes of six core analysis procedures, including SSR mining, sequence clustering, sequence modification, enrichment containing polymorphic SSR sequences, false-positive removal and results output and multiple sequence alignment. After running of transcriptome sequences on this software, a mass of polymorphic SSRs can be identified. The validation experiments showed almost all markers (>90%) that were identified by the SSREnricher as putative polymorphic markers were indeed polymorphic. The frequency of polymorphic SSRs identified by SSREnricher was significantly higher ($P < 0.05$) than that of traditional and HTS approaches. The software package is publicly accessible on GitHub (https://github.com/byemaxx/SSREnricher).

## INTRODUCTION

Microsatellites or simple sequence repeats (SSRs) are tandem repeats of one to six nucleotides present in all eukaryotic genomes. Due to the high mutation rates, co-dominant inheritance, high abundance, reproducibility of microsatellite-based assays, extensive genome coverage, chromosome specific location, amenability to automation and high throughput genotyping (*Parida et al., 2009*), SSRs have emerged as one of the most popular genetic markers in a wide range of applications in population genetics (*Filippi et al., 2015*; *Touma et al., 2019*), conservation biology (*Han & An, 2009*; *Kalia et al., 2011*; *Longwu et al., 2012*), marker-assisted selection (MAS) (*Parida et al., 2009*; *Che et al., 2014*; *Lv et al., 2014*) and other studies (*Moore et al., 1999*; *Palmer et al., 2013*; *Parida et al., 2009*). In addition, there is increasing evidence that SSRs can serve a functional role

Corresponding author
Zongjun Du, 14364@sicau.edu.cn

in the regulation of gene expression by affecting transcription, translational activity, DNA structure, and other metabolic activities (*Palmer et al., 2013*).

Despite the broad applicability of SSRs in animal and plant genetics, their development as genetic markers remains challenging in many species, especially in non-model organisms. The main reason is that they need to be developed de novo in most species being examined for the first time (*Kalia et al., 2011*). Traditionally, microsatellite marker development requires the construction of a genomic library enriched for repeated motifs, isolation and sequencing of microsatellite-containing clones, primer design, optimization of PCR amplification for each primer pair, and a polymorphism assessment in a few unrelated individuals (*Squirrell et al., 2003*). PCR-based isolation of microsatellite arrays (PIMA) (*Lin & Chang, 2013*) and fast isolation by AFLP of sequences containing repeat sites (FIASCO) (*Hakki & Akkaya, 2000*) have been devised; however, most of these techniques are inefficient, time-consuming, multifarious, and expensive (*Abdelkrim et al., 2009*). A survey of service providers indicated that high expense (approximately 5,000–10,000 USD) will deliver approximately 10 polymorphic loci from anywhere between one to more than 3 months by traditional cloning approaches (*Abdelkrim et al., 2009*).

With the wide application of high-throughput sequencing (HTS) technologies, especially for transcriptome sequencing, the development of SSR markers using HTS-based approaches has become a feasible alternative for many species (*Touma et al., 2019*). This method has dramatically reduced the time and cost requirement for large-scale development of SSR markers compared to traditional methods (*Wu et al., 2014*). Nevertheless, the frequency of polymorphic SSR markers developed by this method has also been very low thus far (*Che et al., 2014*; *Lv et al., 2014*; *Yang, Gao & Liu, 2014*), as most of the loci cannot be effectively applied in genetic analyses and marker-assisted selective breeding. To our best knowledge, there are no reports addressing the low frequency of polymorphic SSR marker development.

In this study, we designed a pipeline and integrated it into a user-friendly software called SSREnricher, which can be used to identify polymorphic microsatellites on a large-scale using Trinity (*Grabherr et al., 2011*) assembled transcript files.

## MATERIALS AND METHODS

### Architectural structure

SSREnricher was developed in Python. Polymorphic SSRs enrichment was performed by mining SSR information in transcripts combined with transcript clustering information (Fig. 1). MISA (*Thiel et al., 2003*) was integrated into the software to mine the SSRs in the transcripts, and CD-HIT (*Li & Godzik, 2006*) was used to cluster the transcripts. The program composes of six core analysis procedures, including those outlined in Fig. 2.

1. *SSR mining*: MISA is integrated into SSREnricher and is used to mine SSR sequence information in transcripts, using the following settings: SSR motifs and the number of repeats shown respectively, monomer-10, dimer-6, trimer-5, tetramer-5, pentamer-5, and hexamer-5. The maximal number of interrupting basepairs in a compound microsatellite is set to 100.

2. *Sequence clustering*: CD-HIT-EST is used to cluster transcripts with ≥90% similarity to obtain clustering and positive or negative strand information of sequences.

3. *Sequence modification*: The negative-strand sequences are converted into reverse complementary sequences.

4. *Enrichment containing polymorphic SSR sequences*: The repeat units possessing only one nucleotide are filtered out. Repeat units less than 50 bp from the beginning or end of the sequence are also removed.

5. *False-positive removal*: The software screens for false-positive based on the enrichment results. The clusters that contain sequences with multiple SSRs loci with the same motif, while others miss some SSRs loci, are removed.

6. *Results output and multiple sequence alignment*: Multi-sequence alignment is performed on the obtained clusters to observe the position of the SSRs and generates the result files.

The software is supported on macOS and Linux, and is available in two different versions; one with a graphical user interface allowing for easy use, and one in command-line version, allowing for a more flexible and efficient use through the server.

## Identification of polymorphic SSRs using SSREnricher

To validate our software, we used it to develop SSR markers from rice (*Oryza sativa* L.) and grass carp (*Ctenopharyngodon idella*), the transcriptomes of which are publically available. Analysis in these two species allowed for the validation of our pipeline in both a plant and animal species.

The transcriptomes of rice and grass carp were acquired from the sequence read archive (SRA) database of the National Center for Biotechnology Information (NCBI). Three rice (SRR1799209, SRR1974265, and SRR2048540) and two carp (SRR1618540, SRR1618542) transcriptomes obtained from different ecotypes were acquired. The transcriptomes were also assembled using Trinity.

## Validation experiments

Ten individuals of grass carp and rice were used for SSR validation. For carp, genomic DNA was isolated from tail fin using a salt-extraction method with slight modifications. For rice, genomic DNA was isolated from leaves using the DNeasy Plant mini prep kit (Qiagen, Hilden, Germany). DNA concentration for each sample was determined using NanoDrop (Thermo Scientific, Waltham, MA, USA). Primers were designed using Primer premier 5 software (*Lalitha, 2000*). All the PCR amplifications were carried out in a 10 μL volume containing 1 μL 10× buffer (with $Mg^{2+}$), 100 μM dNTPs, 0.5 μL primer pairs, 1 U Taq DNA polymerase and 20 ng genomic DNA. The reaction program was 5 min at 95 °C, followed by 30 cycles of 30 s at 94 °C, 30 s at annealing temperature 57 °C, 30 s at 72 °C and a final extension at 72 °C for 8 min; then stored at 4 °C. The PCR products were separated by size with Genescan-400 HD size ladders (Applied Biosystems, Foster City, CA, USA), by capillary gel electrophoresis using the ABI 3730 Genetic Analyser (Applied Biosystems, Foster City, CA, USA). The peak heights and fragment sizes were analyzed using GeneMarker software (*Holland & Parson, 2011*).

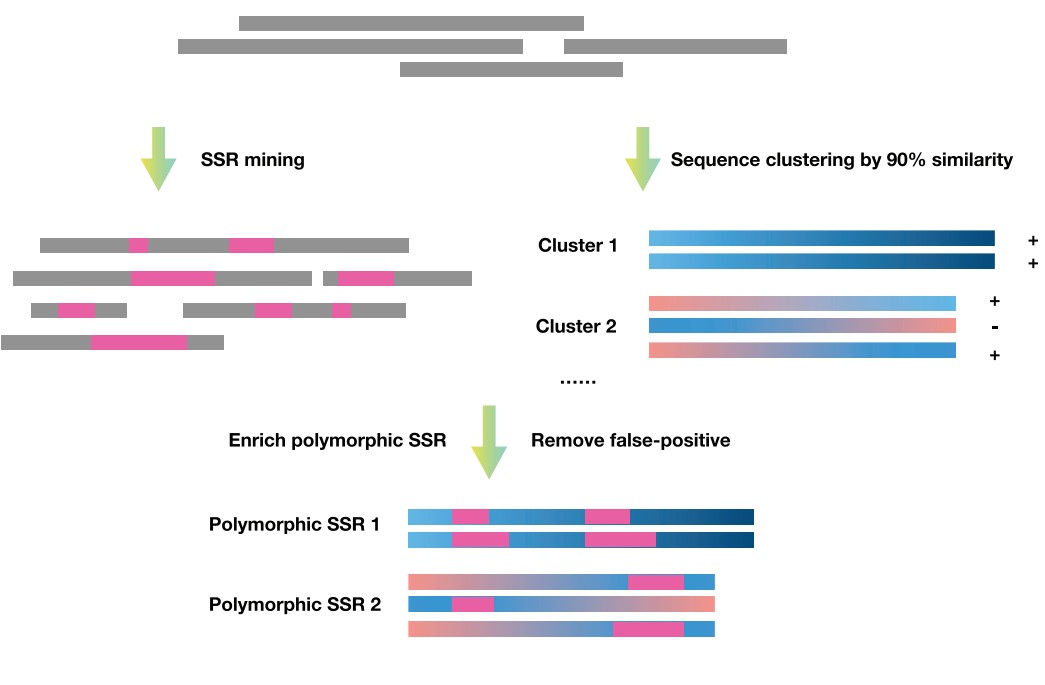

**Figure 1  Schematic diagram of the principles of SSREnricher.** The program first performs SSR mining and sequence clustering on the sequences, and then combines the two results to enrich the polymorphic SSRs.

## RESULTS

Three rice and two carp transcriptomes were used to enrich for polymorphic SSRs. The total number of SSRs identified by MISA (non-including mononucleotides) was 9,901 in rice and 16,700 in carp (Table 1). After enrichment using our pipeline, a total of 480 and 680 putatively polymorphic SSRs were obtained, respectively. The putative polymorphic SSRs enriched by this program were showed in Appendix File 1. Thirty of rice and 20 loci of grass carp were randomly selected for primer design to test the polymorphism of potential SSRs. After PCR amplification and agarose electrophoresis, 27 (90%) and 20 (100%) of the primers could amplify the sequence with distinct product bands in rice and grass carp, respectively. Of which, 26 (96.30%) and 19 (95.00%) loci showed to be polymorphic. The detail information of polymorphic SSRs validated by experiments was showed in Appendix File 2.

We compared the HTS approach (*Wang et al., 2011*) and our pipeline for polymorphic SSRs detection using different transcriptomes in grass carp and rice. The ratio of polymorphic SSRs was raised more than three times. The highest frequency of polymorphic SSRs developed using genomic sequences in rice was 73.6% (*Zheng et al., 2008*), while our pipeline provided a frequency of up to 95%.

We randomly recorded the polymorphic marker frequency in different species, as determined by different methods, including the traditional approach and HTS. The frequency of polymorphic SSRs identified by the traditional clone approach ranged from 5.92% (Estuarine tapertail anchovy (*Coilia nasus*)) to 30.00% (*Pseudogyrincheilus*

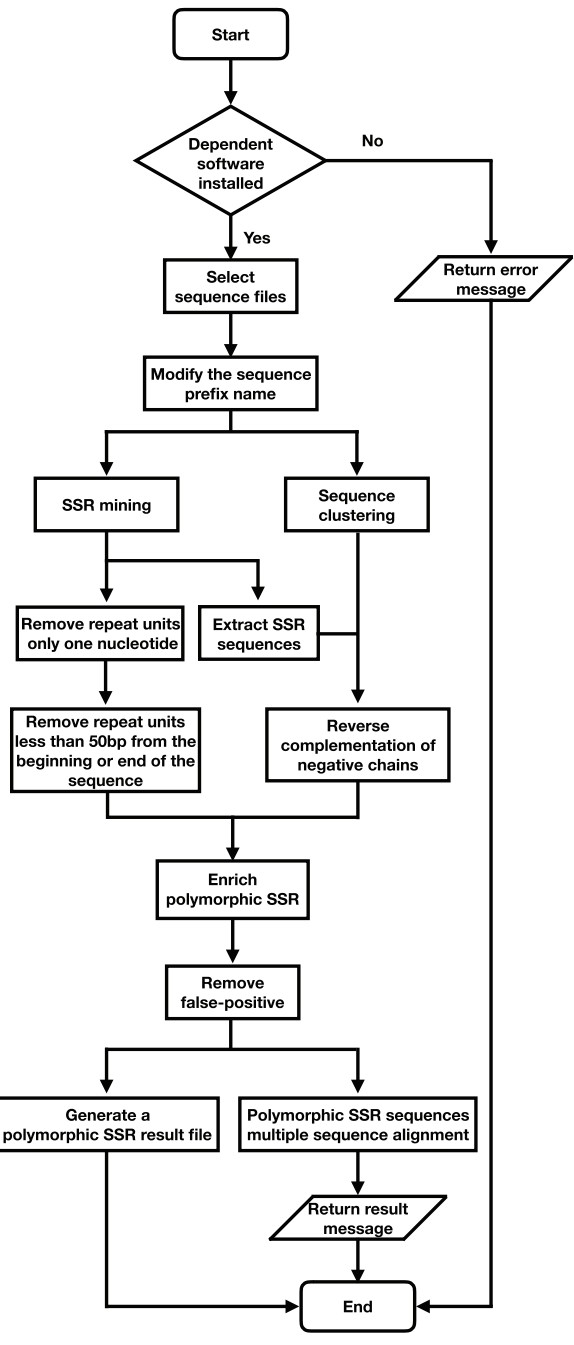

**Figure 2 Program flow chart of SSREnricher.**

*prochilus*) with a mean of 12.04%, while the frequency of polymorphic SSRs divided from transcriptome sequenced by HTS ranged from 9.20% (Pigeonpea (*Cajanus cajan*)) to 73.60% (Rice (*Oryza sativa*)) with a mean of 37.93%. The validation experiment showed that the frequency of polymorphic SSRs identified by SSREnricher was above 90% for both species, with a mean of 93.93%. One-way ANOVA indicated that the frequency of polymorphic SSRs identified by this method was significantly higher than that identified by other methods ($P < 0.05$) (Fig. 3).

**Table 1 Number of SSRs, polymorphic SSRs frequency of two verified species.**

| | Species for verification | |
| --- | --- | --- |
| | Rice | Grass carp |
| Source of the transcriptome | SRR1799209 | SRR1618540 |
| | SRR1974265 | SRR1618542 |
| | SRR2048540 | |
| Total number of SSRs in transcriptome[1] (detected by MISA) | 9,901 | 16,700 |
| Number of putatively polymorphic SSRs enriched by this program | 480[2] | 680[2] |
| Number of primers designed for validation experiments | 30 | 20 |
| Number of primers amplified with clear product bands[3] (%) | 27 (90%) | 20 (100%) |
| Number of polymorphic SSRs[4] (%) | 26[5] (96.30%) | 19[5] (95.00%) |

**Notes:**
[1] The mean number of SSRs in the transcriptome (except mononucleotide) SSRs.
[2] The putative polymorphic SSRs enriched by this program were showed in Appendix File 1.
[3] The percentage of primers that can be amplified with clear products bands.
[4] The percentage of SSRs that are polymorphic.
[5] The detail information of polymorphic SSRs validated by experiments was showed in Appendix File 2.

## DISCUSSION

As the previous reports, the frequency of polymorphic SSR marker development by traditional methods and HTS approaches is relatively low (usually, the frequency ranges from 10% to 70%). Thus, these approaches require a significant amount of time and money for primer design and experimental validation by identifying and separating the many "monomorphic" SSRs from the minority of the polymorphic ones (*Kalia et al., 2011*; *Tang et al., 2008*). In the present study, we developed a software, SSREnricher, which can significantly increase the frequency of polymorphic SSR markers. As demonstrated by the validation experiments in rice and grass carp, the frequency of polymorphic SSR was increased to >90%.

SSREnricher was developed based on the idea of enriching homologous sequences containing the same SSR with a different number of repeats and identifying polymorphic SSRs using computational and bioinformatics analyses. Compared with traditional methods, SSREnricher have eliminated the most intensive wet lab steps in a computational way, significantly reducing the running time. In addition, the cost of our method remains much lower than that of conventional methods. Similar to HTS, the most costly step of SSR marker development is the de novo transcriptome sequencing when there is no available online sequence data in public databases.

The factors that affect the frequency of polymorphic SSRs enriched by the SSREnricher are: (1) genetic differentiation between samples. In theory, the larger genetic differentiation between samples detected by this program, the more putative polymorphic SSRs will be enriched. Thus, it is best not to use individuals with very close relationships, such as full- and half-sib families, to enrich polymorphic SSRs using this program. (2) The specificity of the species itself. The frequency of identifying polymorphic SSRs depends on the species and the genotypes used for the evaluation (*Tang et al., 2008*). According to

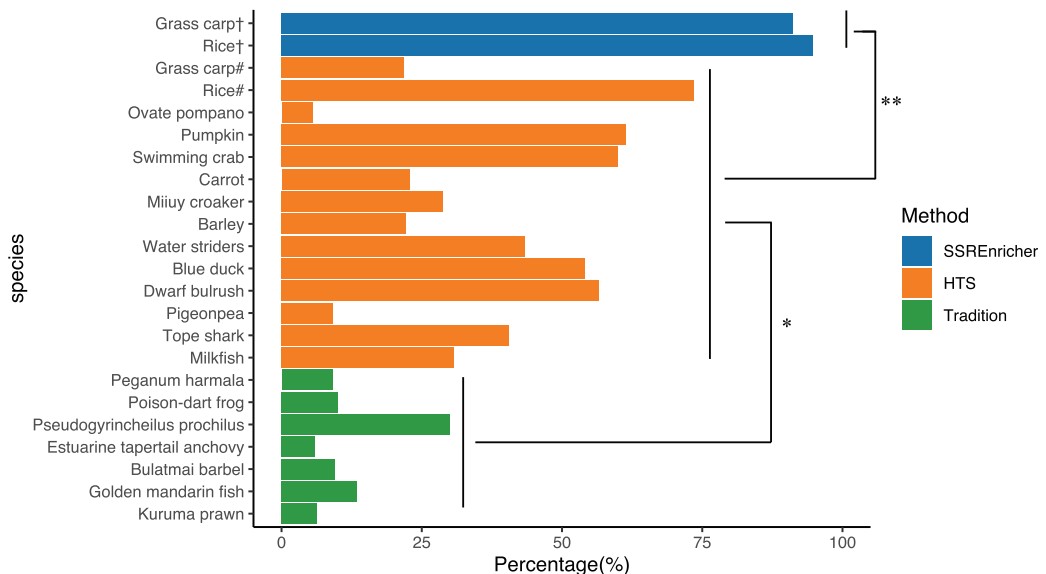

**Figure 3 Comparison of polymorphic marker frequency developed by different methods.** # represents SSRs developed by HTS; † represents SSRs developed by this method; * and ** represents significance with $P < 0.05$ and $P < 0.01$ by ANOVA, respectively. The organisms along with their ratio of polymorphic SSR markers used in this figure were as follows: Kuruma prawn (*Penaeus japonicas*) (*Moore et al., 1999*), Golden mandarin fish (*Siniperca scherzeri*) (*Yang et al., 2012*), Bulatmai barbel (*Luciobarbus capito*) (*Longwu et al., 2012*), Estuarine tapertail anchovy (*Coilia nasus*) (*Yang, Gao & Liu, 2014*), *Pseudogyrincheilus prochilus* (*Shi et al., 2009*), Poison-dart frog (*Dendrobates pumilio*) (*Wang & Summers, 2009*), *Peganum harmala* (*Han & An, 2009*), Milkfish (*Chanos chanos*) (*Santos et al., 2015*), Dwarf bulrush (*Typha minima*) (*Csencsics, Brodbeck & Holderegger, 2010*), Tope shark (*Galeorhinus galeus*) (*Chabot & Nigenda, 2011*), Pigeonpea (*Cajanus cajan*) (*Dutta et al., 2011*), Water striders (*Gerris remigis*) (*Perry & Rowe, 2011*), Barley (*Hordeum vulgare*) (*Thiel et al., 2003*), Miiuy croaker (*Miichthys miiuy*) (*Che et al., 2014*), Carrot (*Daucus carota*) (*Iorizzo et al., 2011*), Pumpkin (*Cucurbita moschata*) (*Wu et al., 2014*), Ovate pompano (*Trachinotus ovatus*) (*Xie et al., 2014*), Swimming crab (*Portunus trituberculatus*) (*Lv et al., 2014*), Blue duck (*Hymenolaimus malacorhynchos*) (*Abdelkrim et al., 2009*), Rice (*Oryza sativa*) (*Zheng et al., 2008*), Grass carp (*Ctenopharyngodon idella*) (*Wang et al., 2011*).

a survey of SSR development using expressed sequence tags (ESTs) in more than fifty species, the frequency of polymorphic SSRs varies dramatically (*Wu et al., 2014*). As previously reported, microsatellite genesis is an evolutionarily dynamic process and has proven to be exceedingly complex, and any change in SSRs resulting in increase or decrease in repeat numbers is associated with mutation rate (*Ellegren, 2004*; *Pearson, Edamura & Cleary, 2005*).

With the rapid development of sequencing technologies, the cost and time have been greatly reduced, allowing for widespread use of high-throughput sequencing methods. The number of sequenced and deposited transcriptomes in public databases for many economically important species is continuously increasing. As a consequence, the large-scale development of SSR markers at a low cost and within a short timescale is becoming a feasible task. Thus, development of polymorphic SSR markers using SSREnricher has attractive potential applications in the fields of genetic analyses and molecular selective breeding in plants and animals.

## CONCLUSIONS

Almost all tested SSRs predicted to be polymorphic by the SSREnricher are indeed polymorphic, indicating development of polymorphic SSR markers by the SSREnricher is efficient and reliable. The frequency of polymorphic SSR markers can be significantly increased, and thus the time and cost can be notably decreased by using the SSREnricher.

### Funding

This study was funded by the National Natural Science Foundation of China (No. 41706171) and the 13th five-year aquaculture-breeding project of Sichuan province (No. 2016NYZ0047), China. The funders had no role in study design, data collection and analysis, decision to publish, or preparation of the manuscript.

### Grant Disclosures

The following grant information was disclosed by the authors:
National Natural Science Foundation of China: 41706171.
13th Five-Year Aquaculture-Breeding Project: 2016NYZ0047.

### Competing Interests

The authors declare that they have no competing interests.

### Author Contributions

- Wei Luo conceived and designed the experiments, analyzed the data, prepared figures and/or tables, authored or reviewed drafts of the paper, and approved the final draft.
- Qing Wu conceived and designed the experiments, analyzed the data, prepared figures and/or tables, authored or reviewed drafts of the paper, and approved the final draft.
- Lan Yang performed the experiments, prepared figures and/or tables, and approved the final draft.
- Pengyu Chen performed the experiments, prepared figures and/or tables, and approved the final draft.
- Siqi Yang performed the experiments, prepared figures and/or tables, and approved the final draft.
- Tianzhu Wang performed the experiments, prepared figures and/or tables, and approved the final draft.
- Yan Wang performed the experiments, prepared figures and/or tables, and approved the final draft.
- Zongjun Du conceived and designed the experiments, authored or reviewed drafts of the paper, and approved the final draft.

### Data Availability

Data is available at GitHub: https://github.com/byemaxx/SSREnricher.

## Supplemental Information

Supplemental information for this article can be found online at http://dx.doi.org/10.7717/peerj.9372#supplemental-information.

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
