# Peer review of "SSREnricher: a computational approach for large-scale identification of polymorphic microsatellites based on comparative transcriptome analysis"

_PeerJ, doi:10.7717/peerj.9372_

## Round 0.1 · original submission · Major Revisions

Dear Dr. Luo and colleagues:

Thanks for submitting your manuscript to PeerJ. I have now received two independent reviews of your work, and as you will see, the reviewers raised some concerns about the research (mostly the manuscript format and content). Despite this, these reviewers are optimistic about your work and the potential impact it will have on research studying informatics analysis of transcript-derived microsatellite data. Thus, I encourage you to revise your manuscript, accordingly, taking into account all of the concerns raised by both reviewers.

Please have an English-speaking expert help revise your manuscript. Focus on clarity, and address the sections considered incomplete by the reviewers. It appears that certain key references are missing. The Methods should be clear, concise and repeatable. Please ensure this. Also, elaborate on the discussion of your findings, placing them within a broad and inclusive body of work by the field.

I look forward to seeing your revision, and thanks again for submitting your work to PeerJ.

Good luck with your revision,

-joe

Reviewer 1 ·

Basic reporting

The authors illustrate the development of a new software called SSREnricher to enrich polymorphic SSRs through the comparative transcriptome analysis. The issue is quite interesting and certainly of considerable applicability, reducing the lobariosity of traditional metodologies for microsatellite identification within species. Anyway, it is not clear whether this software can also work using genomic data or it is limited to EST-SSR.
The manuscript is written in an extremely simple and clear English language, although sentences without grammatical construction or too short often recur. Bibliography as well as the scientific discussions should be enriched.
The article structure should be modified with “Results and discussion” and “Conclusions” instead of “Results” and “Discussion and conclusions”. Figures and supplementary files should be better labeled and accompanied by more detailed captions.

Experimental design

The research is in accordance with the aims of the journal. However, in my opinion, it has been described in a too reductive way, because the potential of this software with genomic data is missing, for example, as well as the description of the performed wet experiments.

Validity of the findings

The research is very interesting and of considerable application impact, thus meeting the needs of scientific community. Anyway, not always well described or in a detailed form. The manuscript needs major revisions and some aspects to be clarified before pubblication.

Additional comments

Introduction:
raws 44-47 and 60-61: please, add some important references.
Mat&Met:
The verbs in this section are usually in a past tense. Furthermore, since the research include bioinformatic analysis and validation by wet experiments, this section should be enriched with protocols and details of the lab work done (sample preparation, DNA extraction, PCR reaction, agarose gel and capillary electrophoresis). If they refer to already known protocols, they should be mentioned as references. Moreover, punctuation must be corrected and short sentences without grammatical construct must be avoided (ex. raws 75-77).
Raw 86: please, specify the meaning of “c1” and “p”.
Raw 87: this sentence is not clear. Perhaps authors mean that SSRs less than 50 bp long have been eliminated? Please clarify.
Raw 88: replace “cluster” with “clusters”.
Results:
95: I would suggest “Results and discussion”.
96: please, provide the scientific names of both species here and remove them from raws 101-102.
98: “ecotypes”? Or genotypes? Please clarify. Moreover, replace "with" with “obtained from”.
103-106: “Primers were… .. software”. These sentences should be moved to Mat&Met. Moreover, since real experiments have been carried out (DNA extraction, PCR, agarose gel, capillary electrophoresis), all information about that must be added in M&M section in detail.
107: It would be important to indicate the number of rice and carp samples used to validate the identified SSRs and how many polymorphic alleles were found per each SSR. I guess the supplementary "primer" table could be actually included in the main text, with a detailed caption. A figure illustrating polymorphic peaks of some selected SSRs would also be useful.
I suggest to include the discussion in this section, rather than “discussion and conclusions”. Anyway, the scientific discussion is actually poor and of course needs to be extended.
Discussion and conclusions:
115: please replace with "Conclusions".
Figures:
Fig.1 needs a more descriptive caption.
Fig.3: Please, clarify the meaning of “traditional” in caption and remove all the spaces between brackets and commas.
Supplementary material:
The main text of the manuscript is missing about any reference to the additional files, as well as the corresponding captions, which anyway must be detailed and must also clarify the presence of symbols in the table itself (e.g. what do they mean by N in primer tab? The number of alleles?).
Moreover, locus name must be corrected with Os and Ci, respectively, instead of Orsa and Ctid (e.g. Os-SSR1 and Ci-SSR2 and the same for all loci). As already mentioned, the primer tab could be included in the main text.
Furthermore, header is completely missing in the polymorphic_SSR tab; what does each column of the table indicate?

Reviewer 2 ·

Basic reporting

The article is not clear, the language is not professional and there are several mistakes in the text.
The references are too old and some important referenceses are totally missing.

The article is too short and several steps and descriptions are not present.

The raw data are correctly shared.

The results are supported by the experimental work.

Experimental design

The aim of the article is appropriate to the scope of the journal.

The reserach question is well defined but not efficiently supported.

The investigation is not so rigorous.

Methods are not well described.

Validity of the findings

All raw data and primer sequences are provided, but is not well explained the real impact of the research. The discussion is not a real discussion, because is totally missing a comparison with other similar tools.

Additional comments

Dear authors, in my opinion the manuscript needs major revisons.
The introduction contains references too olds, and it is incomplete.
The materials and methods are not well explained, the validation of SSR si totally missing.
The results section is confused and reports some steps of methods.
The discussion is not a discussion: there are no comparisons with others similar bioinformatic tools.
In my opinion there is a large part of work that is missing, and the article cannot be published in this form.

---

## Round 0.2 · Major Revisions

Dear Dr. Luo and colleagues:

Thanks for revising your manuscript based on the concerns raised by the reviewers. I have now received feedback from the two original reviewers of your work, and as you will see, one is more favorable, the other less so; so, more needs to be done. The reviewers raised some new (and unaddressed) concerns about the research, and areas where the manuscript can be improved. Please consider missing and outdated references, as well as issues related to sampling.

I agree with the concerns of the reviewers, and thus feel that their suggestions should be adequately addressed before moving forward. Please note that Reviewer 1 kindly provided a marked-up version of your manuscript.

Therefore, I am recommending that you revise your manuscript, accordingly, taking into account all of the issues raised by the reviewers. I do believe that your manuscript will be closer to publication quality once these issues are addressed.

Good luck with your revision,

-joe

Reviewer 1 ·

Basic reporting

The authors provided an improved version of their manuscript compared to the first one, answering almost all my requests (someone is still open). The addressed issue is certainly useful, English language is definitely improved, bibliography and the scientific discussions have been enriched. Anyway, the manuscript still needs some features to be assess and some minor corrections, before publication. For instance:
1) How many carp and rice samples did authors test for SSR validation? The sample size is relevant for SSR polymorphisms and discrimination power.
2) In accordance to the sample size, the number of SSRs selected for wet validation takes on importance. Now it represents the 4% of the computationally identified SSR in both species.
3) Authors ignored the hypothesis to add a picture of the identified polymorphic profiles (alleles by capillary electrophoresis).
4) The references to supplementary materials are missing in the main text.
5) Please correct le abbreviation of specie Latin names.
My detailed comments are on the pdf file.

Experimental design

The research is in accordance with the aims of the journal. In this new version the experimental design has been deeply described and became reproducible.

Validity of the findings

The research has high potential with regard to applicability and needs of the scientific community.

Additional comments

Abstract:
raw 31: replace semicolon with a dot.

Introduction:
raws 44-46: single o multiple references must be added following exactely the refer research, i.e: “SSRs have emerged as one of the most popular genetic markers in a wide range of applications in population genetics (specific ref), conservation biology (specific ref), and marker-assisted selection (MAS) (new ref).” Please correct and add proper references.

Mat&Met:
raw 31: replace “the developed” with “our”.
raw 104: replace "with" with “obtained from”.
raws 106-107: I would remove this sentence since it’s a repetition.
raws 106-109: I still believe that the number of used rice and carp samples for SSR validation could be of considerable importance.

Results:
Table 1: please check apics.
Fig.3: please provide ANOVA results in both captation and graph.
Moreover, a figure illustrating polymorphic peaks of some selected SSRs should be useful.

Supplementary material:
The main text of the manuscript is still missing about any reference to the additional files. Moreover, locus name have not been corrected jet according to the conventional scientific language (Oriza sativa is shorten as Os!).

Annotated reviews are not available for download in order to protect the identity of reviewers who chose to remain anonymous.

Reviewer 2 ·

Basic reporting

The article is improved, however the large part of the citations is too old. The introduction and discussion are not sufficient. The added table 1 is not suitable for a scientific publication (i.e. number instead of N or No)

Experimental design

Also if material and methods were improved the experimental design is not clear and does not fit with the discussion.

Validity of the findings

The discussion session is poor and does not fit with the experimental work:
In material and methods you report:
"To validate the developed software, we used it to develop SSR markers in rice (Oryza sativa L.) and in grass carp (Ctenopharyngodon idella), the transcriptomes of which are publically available. Analysis in these two species allowed for the validation of our pipeline in both a plant and animal species."

In the discussion you write:

"The frequency of polymorphic SSR marker development by traditional methods and HTS approaches is relatively low (usually, the frequency ranges between 10% and 50%), especially in species with low polymorphism, such as Pigeonpea (Cajanus cajan) and ovate pompano (Trachinotus ovatus)(Dutta et al., 2011; Xie et al., 2014)."

What does it mean? I need to have a comparison and discussion with other systems validate in rice and grass carp.

Additional comments

Dear authors I tried to give you an opportunity to publish this article, but unfortunately this revised version is too far from a corrected version suitable for publishing.

---

## Round 0.3 · accepted · Accept

Dear Dr. Luo and colleagues:

Thanks for revising your manuscript based on the concerns raised by the reviewers. I now believe that your manuscript is suitable for publication. Congratulations! I look forward to seeing this work in print, and I anticipate it being an important resource for groups studying informatics analysis of transcript-derived microsatellite data. Thanks again for choosing PeerJ to publish such important work.

Best,

-joe